# Bovine Serum Albumin Molecularly Imprinted Electrochemical Sensors Modified by Carboxylated Multi-Walled Carbon Nanotubes/CaAlg Hydrogels

**DOI:** 10.3390/gels9080673

**Published:** 2023-08-20

**Authors:** Letian Cheng, Zhilong Guo, Yuansheng Lin, Xiujuan Wei, Kongyin Zhao, Zhengchun Yang

**Affiliations:** 1State Key Laboratory of Separation Membranes and Membrane Processes, Tiangong University, Tianjin 300387, China; 2010210601@tiangong.edu.cn (L.C.); zhilong_guo@tjeminent.com (Z.G.); 2131020309@tiangong.edu.cn (Y.L.); 1710210129@tiangong.edu.cn (X.W.); 2Tianjin Key Laboratory of Film Electronic & Communication Devices, Tianjin University of Technology, Tianjin 300384, China; yangzhengchuntjut@163.com

**Keywords:** calcium alginate hydrogel, protein molecular imprinting, electrochemical sensor, carboxylated multi-walled carbon nanotubes

## Abstract

In this paper, sodium alginate (NaAlg) was used as functional monomers, bovine serum albumin (BSA) was used as template molecules, and calcium chloride (CaCl_2_) aqueous solution was used as a cross-linking agent to prepare BSA molecularly imprinted carboxylated multi-wall carbon nanotubes (CMWCNT)/CaAlg hydrogel films (MIPs) and non-imprinted hydrogel films (NIPs). The adsorption capacity of the MIP film for BSA was 27.23 mg/g and the imprinting efficiency was 2.73. The MIP and NIP hydrogel film were loaded on the surface of the printed electrode, and electrochemical performance tests were carried out by electrochemical impedance spectroscopy (EIS) and differential pulse voltammetry (DPV) using the electrochemical workstation. The loaded MIP film and NIP film effectively improved the electrochemical signal of the bare carbon electrode. When the pH value of the Tris HCl elution solution was 7.4, the elution time was 15 min and the adsorption time was 15 min, and the peak currents of MIP-modified electrodes and NIP-modified electrodes reached their maximum values. There was a specific interaction between MIP-modified electrodes and BSA, exhibiting specific recognition for BSA. In addition, the MIP-modified electrodes had good anti-interference, reusability, stability, and reproducibility. The detection limit (LOD) was 5.6 × 10^−6^ mg mL^−1^.

## 1. Introduction

Materials that specifically recognize proteins have a wide range of promising applications in the fields of biosensing [1], diagnostic analysis [2], proteomics [3], and controlled drug release [4]. Although natural antibodies were widely used, they suffered from high production costs, being time-consuming, and having poor stability and reproducibility [5,6]. Molecular imprinting technology is a technique to prepare polymers (molecular imprinting polymers, MIPs) with specific recognition properties for target molecules (template molecules), which combines the affinity and specificity of antibodies with the durability of synthetic materials [7,8]. However, relative to the blotting of small molecule templates, the imprinting of protein molecules encounters great challenges because proteins are variable in structure, large in size, and contain a large number of active functional groups, making it difficult to obtain the precise pore and binding sites [9,10]. MIPs with precise recognition of pores [11] and binding sites by proteins must simultaneously satisfy the following two conditions: (1) the conformation of the protein is maintained during the self-assembly and polymerization reaction of the protein and monomer [12] and (2) the protein is sufficiently removed from the polymer without destroying the structure of the imprinted pores [13].

However, monomers commonly used in the literature such as acrylamide, methacrylic acid, and acrylic acid disrupted the structure of proteins [14]. Taking bovine serum albumin (BSA), the most used template for protein molecular blotting, as an example, the secondary structure of BSA was significantly changed, even with a 200:1 molar ratio of AM to BSA, such that a 350:1 ratio resulted in a 50% reduction in α-helix content. Because small molecules of functional monomers easily penetrate inside the protein and break the hydrogen bonds that maintain a stable protein structure, the molar ratio of functional monomers to protein in the preparation of MIP usually exceeds 1000:1. So, it obtained the imprinted pore of the protein after structural alteration or even denaturation [14]. Qian et al. proposed a method to imprint proteins using macromolecular monomers [15]. The results of circular dichroism and simultaneous fluorescence spectroscopy showed that macromolecular monomers stabilized the structure of proteins, and the obtained MIPs had higher imprinting efficiency and recognition ability than those prepared from small molecular monomers [15].

The use of surface imprinting facilitated the elution and recombination of proteins. Liu et al. prepared glycoprotein molecularly imprinted polymers conveniently and efficiently by controlled directed surface imprinting [16]. Zhang et al. prepared the heat-sensitive surface-imprinted nanoparticles that could specifically capture and release targeted proteins from human plasma [17]. Using polydopamine-like mussel mucin materials, Qin et al. constructed surface molecularly imprinted polymer-based sensors to achieve highly sensitive and selective rapid detection of proteins [18]. Shi et al. deposited template proteins adsorbed on mica sheets [19], and then disaccharides were encapsulated on the adsorbed protein surface. They used epoxy resin to immobilize the film. Finally, the mica sheet was peeled off, leaving pores complementary to the template. Surface imprinting technology had become the main strategy for macromolecular imprinting, which was especially suitable for the field of sensors. Protein molecularly imprinted electrochemical sensors had the advantages of good selectivity, high sensitivity, a low detection limit, reusability, and easy preparation [20].

Our group prepared a series of protein molecularly imprinted hydrogels using sodium alginate as a macromolecular monomer, which adequately maintained the conformation of proteins [21]. However, the calcium alginate (CaAlg) hydrogels had low strength, thus making it difficult to maintain the imprinted pores. Double network hydrogels containing chemically cross-linked and physically cross-linked structures, such as polyacrylamide/calcium alginate hydrogels, have both high strength and toughness [22]. Our group prepared BSA-imprinted polyacrylamide/calcium alginate hydrogel films with good recognition selectivity for template proteins, even when the cross-linking agent was six parts per million of the monomer mass [23]. However, the calcium ions in the hydrogel were easily replaced by monovalent cations, leading to swelling and disrupting the structure of the imprinted pores. In this paper, bovine serum albumin (BSA) was used as the template molecule; carboxylated multi-walled carbon nanotubes (CMWCNTs) and sodium alginate (NaAlg) were co-dissolved in water and cross-linked by calcium ions to prepare BSA molecularly imprinted CMWCNT/CaAlg hydrogel films. The BSA molecularly imprinted carboxylated multi-walled carbon nanotube/CaAlg hydrogel film was modified onto the screen-printed carbon electrode surface to make a molecularly imprinted electrochemical sensor, which specifically adsorbed and recognized BSA molecules through the electrochemical workstation output signal.

## 2. Results and Discussion

### 2.1. Characterizations of BSA Molecularly Imprinted CMWCNT/CaAlg Hydrogel Films

Figure 1 shows the SEM images of the CaAlg hydrogel film and the BSA-imprinted CMWCNT/CaAlg hydrogel film without eluting and after eluting. As shown in Figure 1, the surface of the CaAlg hydrogel film was relatively smooth. The surface of the uneluted CMWCNT/CaAlg hydrogel film was smoother than the eluted hydrogel film. The main reason was that when the template protein BSA was removed from the MIP hydrogel film, the imprinted holes were left, which increased the surface roughness of the hydrogel film.

Figure 2 shows the swelling rate of the CMWCNT/CaAlg hydrogel film in normal saline at different times and the equilibrium swelling rate of the films with different CMWCNT contents. As shown in Figure 2a, the swelling rate of all the samples exhibited a pattern of first increasing and then flattening. The swelling rate increased significantly within 60 min. With the increase in the content of CMWCNT, the anti-swelling performance of the CMWCNT/CaAlg hydrogel significantly improved. In Figure 2b, it can be seen that the higher the content of CMWCNT, the lower the equilibrium swelling rate. The addition of CMWCNT can effectively enhance the anti-swelling performance of the CaAlg [24]. 

### 2.2. Mechanical Properties of BSA Molecularly Imprinted CMWCNT/CaAlg Hydrogel Films

As shown in Figure 3, the addition of a small amount of CMWCNT significantly improved the mechanical properties of the CaAlg hydrogel film. Figure 3a shows the stress–strain curves of BSA molecularly imprinted CMWCNT/CaAlg hydrogel films with different CMWCNT contents. The tensile strength of the CMWCNT/CaAlg hydrogel film was higher than the CaAlg film. When the CMWCNT content in NaAlg was 2 wt.%, the maximal tensile strength reached a value of 1440 Kpa. The formation of the Ca^2+^ cross-linking synchronously with the COO of CMWCNT and the COO of NaAlg increased the cross-linking density and improved the strength of the CMWCNT/CaAlg hydrogel film. The interactions between CMWCNT and NaAlg were investigated by molecular dynamic (MD) simulation in another paper we published [24]. 

When the content of the CMWCNT was 2 wt.% of NaNAlg, the mechanical properties of the CaAlg hydrogel film were best. At this time, the dispersion of the CMWCNT in the CaAlg matrix had reached saturation. The dispersion effect of the CMWCNT was improved after carboxylation modification, but the improvement was limited. When the content of the CMWCNT exceeded 2 wt.%, the CMWCNT would agglomerate, reducing the mechanical properties of the CaAlg hydrogel film. 

### 2.3. Adsorption Properties of BSA Molecularly Imprinted CMWCNT/CaAlg Hydrogel Films

Figure 4 shows the effect of elution time on the adsorption properties of MIP and NIP. As shown in Figure 4, the longer the elution time, the cleaner the protein was eluted. As the elution time increased, the BSA adsorption capacity of both MIP and NIP films increased. When the elution time reached 300 min, the adsorption capacity of the MIP and NIP film was 27.23 and 10.04 mg/g, respectively. When the elution time was greater than 300 min, both the adsorption capacity and the imprinting efficiency reached an equilibrium state, and the imprinting efficiency was 2.73. 

The pH value of the protein elution solution also had a significant impact on the adsorption performance. Figure 5 shows the adsorption capacity of BSA on MIP and NIP films after elution with different pH eluents. As shown in Figure 5, the optimal pH of Tris HCl eluent was 7.4. When the pH of the eluent was less than 7.4, the template protein was not completely eluted, resulting in a decrease in the imprinted pores and a lower adsorption capacity. However, when the pH of the eluent exceeded 7.4, the eluent would destroy the structure of imprinted holes in the CMWCNT/CaAlg film, leading to a decrease in the adsorption capacity of the MIP film. These results were consistent with the results of our previously published papers [20]. 

### 2.4. Effect of Alginate on BSA Conformational

The macromolecular monomer sodium alginate was used for the preparation of MIP in this paper, and sodium alginate did not cause the denaturation of BSA. For further analysis, the change in BSA secondary structure was studied by circular dichroism (CD) spectroscopy [25]. The CD spectra of pure BSA and BSA eluted with Tris HCl solution are shown in Figure 6. As reported in the literature [26,27,28], the CD spectrum of pure BSA exhibited two negative elliptical peaks (curve a) in the far ultraviolet region at 208 nm and 222 nm, which were the characteristics of α-spiral structures of BSA. After elution with Tris HCl solution at pH = 7.4, the intensity of BSA slightly decreased at 208 and 222 nm [29], but its CD spectral curve did not show significant changes compared to pure BSA, indicating that the BSA secondary structure was not changed during the elution process. It was also verified that the structure of BSA was not damaged during the preparation process. 

### 2.5. Electrochemical Characterization of Different Electrode Surfaces

Figure 7a shows the electrochemical impedance spectroscopy (EIS), where all four curves were composed of semicircles and straight lines. The semicircle diameter represented the electron transfer resistance. The smaller the diameter, the smaller the resistance was. The linear line represented the diffusion-limited part. Compared with the bare carbon electrode, the resistance of electron transfer of the pure CaAlg-modified electrode increased, which was due to the fact that the loaded CaAlg hydrogel blocks the electron transfer on the electrode surface. The resistance of the hydrogel-modified electrode with CMWCNT was relatively reduced because the carboxylated multi-wall carbon nanotubes contained in the film had good conductivity and increased the electron transfer ability. The electron transfer resistance of the MIP-modified electrode after elution was higher than the MIP-modified electrode without elution, which proved the existence of imprinted holes. Figure 7b shows the differential pulse voltammetry (DPV) curves. Compared to other electrodes, the BSA eluted electrode had the highest peak current value. The peak current value of the MIP-modified electrode after elution was higher than the MIP-modified electrode without elution, indicating that the imprinted pores were left after removing the template protein through elution. The imprinted pores provided more channels for [Fe(CN)_6_]^3−^ to diffuse to the surface of the MIP-modified electrode.

### 2.6. Effect of Eluent Times, Adsorption Times, and Eluent pH on DPV Response Currents of the MIP-Modified Electrodes

The BSA template was removed from the CMWCNT/CaAlg film in Tris HCl eluate with pH = 7.4, and the effect of elution time on the DPV response currents of MIP-modified electrodes was investigated. As shown in Figure 8, when the elution time increased, the peak current value showed a trend of first increasing and then flattening. After 15 min of elution, the peak current gradually stabilized, indicating that BSA had been completely eluted at this time. A shorter elution time could ensure the activity of the electrode, and a 15 min elution time was selected. Compared to the MIP film in Figure 5, the elution time of the MIP film modified on the electrode was much shorter because the CMWCNT/CaAlg film on the screen-printed electrode was very thin, with only a few microns. Therefore, BSA could rapidly release from the CMWCNT/CaAlg film-modified electrode in Tris HCl eluate with pH = 7.4 [30].

As shown in Figure 9, when the adsorption time was 0 and no BSA was adsorbed, the peak current value was the highest, with a value of about 0.22 mA. At this time, there was no protein adsorption on the MIP film, and the imprinted pores were used as the electron transfer channels on the electrode. When the adsorption time gradually increased, the peak current value showed a trend of first decreasing and then flattening. The reason was that the template protein entered the imprinted pores on the MIP film and recombined with the imprinting sites, thereby reducing the electron transfer channels on the electrode and weakening the electrochemical signal. After adsorption for 15 min, the peak current value did not change significantly with the change in adsorption time. It indicated that the adsorption of BSA on MIP-modified electrodes had reached equilibrium, so the 15 min adsorption time was better and was selected.

As shown in Figure 10, when the pH value of the eluent increased, the peak current showed a trend of first increasing and then decreasing. When pH < 7.4, as the pH value of the eluent increased, the peak current also increased accordingly. When pH > 7.4, as the pH value of the eluent increased, the peak current value did not increase but decreased. When the pH value was higher than 7.4 and continued to increase, the peak current value of the MIP-modified electrode decreased, because the pH value was too high and the CMWCNT/CaAlg hydrogel film loaded on the electrode surface swelled too much, thus affecting the electrochemical response of the MIP-modified electrode. On the other hand, the change in pH value had almost no effect on the peak current values of NIP-modified electrodes, and the peak current values remained around 0.04 mA. The reason was that the NIP film had no imprinted pores, and after elution with the eluent, the surface and interior of the CMWCNT/CaAlg film did not produce imprinted pores that match the template molecules, which had little impact on the electrochemical signal. When the pH value was 7.4, the peak current value was relatively high, indicating that the conditions for eluting template proteins at this pH were relatively mild and could effectively protect the stability of imprinted holes in the MIP film.

### 2.7. Selective Detection and Imprinting Efficiency of the MIP- and NIP-Modified Electrodes

The results of the selective detection of BSA by MIP-modified electrodes and NIP-modified electrodes were shown in Figure 11. In this experiment, bovine hemoglobin (BHb), ovalbumin (OVA), and lysozyme (Lys) were selected for comparison proteins. In Figure 11 it could be found that after the MIP-modified electrode was combined with BSA, its current change value (ΔI) was significantly higher than other proteins. When the protein was BSA, the BSA could enter the imprinted pores and recombine with the polymer. When the adsorbed protein was not BSA, the protein could not enter the imprinted pores, and its structure, size and shape were significantly different from BSA. Comparing the peak current values of NIP-modified electrodes with other proteins, it is found that there was no significant difference, which was mainly due to the absence of imprinted pores in the NIP-modified electrodes. 

To further validate its selectivity, the imprinting efficiency (IF = ΔI_MIPs_/ΔI_NIPs_) of various proteins was calculated [31]. Among them, ΔI_MIPs_ and ΔI_NIPs_ referred to the response current changes generated by MIP-modified electrodes and NIP-modified electrodes after protein adsorption. As shown in the figure, the IF value of BSA was much higher than other substances, which fully demonstrated the good specificity recognition ability of MIP-modified electrodes for BSA.

### 2.8. Reusability, Stability, and Reproducibility of MIP-Modified Electrodes for BSA Detection

Selectivity is a significant indicator to evaluate the success of MIP electrochemical sensor preparation. As shown in Figure 12a, the peak current value of DPV was obtained from the same MIP electrochemical sensor. It is found that the DPV peak current values measured after five elutions and re-adsorption remain around 0.2 mA, indicating that the MIP-modified electrode has good reusability. 

In Figure 12b, it can be seen that there was no significant change in the electrochemical response of the coating after 12 days, and the coating still maintained an electrochemical response of about 80% after 30 days, indicating that the MIP-modified electrode had good stability because carboxylated multi-walled carbon nanotubes can improve the structural stability of the electrode and fix imprinted pores. 

Figure 12c shows the peak current values of DPV obtained after eluting template molecules with six different MIP-modified electrodes. It can be seen that the DPV peak current values of six different MIP-modified electrodes remain around 0.2 mA. After calculation, the relative standard deviation between different MIP-modified electrodes was about 4.3%, indicating that the MIP-modified electrodes have good reproducibility.

### 2.9. Linear Range, Detection Limit of the Sensor, and Comparison with Other Materials

The main parameters of this sensor were described and addressed, e.g., sensitivity and detection limit. As shown in Appendix A, the favorable linear relationship between BSA concentration and peak current value is in the range of 5.6 × 10^−6^ mg mL^−1^ to 1.2 × 10^−3^ mg mL^−1^, with the linear equation ΔI (μ A) = 0.94C_BSA_ + 4.21. The linear correlation coefficient R^2^ was 0.998 and the detection limit (LOD) was 5.6 × 10^−6^ mg mL^−1^ (S/N = 3). 

A brief comparison study between the results of previous works is in Table 1. It was found that the MIP electrochemical sensor prepared in this work has higher sensitivity and lower detection limits compared with previous reports.

## 3. Conclusions

In this study, CMWCNT was introduced into the CaAlg hydrogel to form a composite hydrogel. The mechanical and anti-swelling properties of the CaAlg hydrogel were improved. The BSA imprinted CMWCNT/CaAlg hydrogel (MIP) film and non-imprinted CMWCNT/CaAlg hydrogel (NIP) film were prepared. The adsorption capacity of the MIP film for BSA was 27.23 mg/g, and the imprinting efficiency reached 2.73. The MIP and NIP hydrogel films were loaded on the surface of the printed electrode to prepare the electrochemical sensor, and electrochemical performance tests were carried out. There was a specific interaction between MIP-modified electrodes and BSA, exhibiting specific recognition for BSA. The MIP-modified electrodes had good anti-interference, reusability, stability, and reproducibility. The detection limit (LOD) was 5.6 × 10^−6^ mg mL^−1^, with a detection range of 5.6 × 10^−6^ mg mL^−1^ to 1.2 × 10^−3^ mg mL^−1^. In addition, the preparation and research method of the electrochemical sensor was simple and used non-toxic and harmless substances, which was convenient for large-scale production.

## 4. Materials and Methods

### 4.1. Materials

The Appendix A contains descriptions of the materials. 

### 4.2. Preparation of BSA Molecularly Imprinted CMWCNT/CaAlg Hydrogel Films

Different concentrations of CMWCNT (0%, 0.5%, 1%, 2%, 3%) were added to each beaker, along with 20 mL of deionized water. To achieve uniform distribution of the CMWCNT in the solution, the beakers were subjected to ultrasonic cleaning for 30–40 min. Next, we added 0.02 g of BSA and 0.52 g of NaAlg to each beaker and stirred the mixture until it reached a homogeneous consistency. The resulting cast solution was then refrigerated to facilitate defoaming. To create the gel film, we used a glass rod with copper wires of varying diameters (0.2 mm, 0.3 mm, 0.4 mm, 0.5 mm) to scrape the solution onto a glass plate. Following the scraping process, the film was cross-linked in a 2.5 wt.% CaCl_2_ solution for 5 min. Subsequently, the gel film was carefully detached from the glass plate and transferred to a container filled with 2.5 wt.% CaCl_2_ solution, where it underwent further cross-linking for a duration of 4 h. This process led to the formation of BSA molecules within the CMWCNT-doped calcium alginate hydrogel films, referred to as MIPs. The preparation of NIP hydrogel films followed a similar procedure to the BSA molecularly imprinted CMWCNT/CaAlg hydrogel films (MIPs), with the exception that BSA, which was not added.

### 4.3. Preparation of BSA Molecularly Imprinted CMWCNT/CaAlg Hydrogel-Modified Electrochemical Sensors

Figure 13 illustrates the experimental procedure for preparing the bare carbon electrode for modification. The electrode underwent a thorough cleansing process, involving washing with distilled water to remove impurities. Subsequently, the electrode was placed in a glass dish and soaked in ethanol for 1 h to further eliminate surface contaminants. Following the ethanol treatment, a 5% volume ratio of KH550 silane, containing amino active groups, was added to the glass dish, enabling the activation of the electrode surface through chemical couplings with the highly nucleophilic amine system present in the substrate. The soaked bare carbon electrodes were then dried in the bake oven and set aside, where it was observed that the hydrogel adhered effectively to the electrode surface. To complete the modification process, the MIP-modified electrode was cross-linked in a 2.5 wt.% CaCl_2_ aqueous solution for 4 h. Afterward, the BSA was eluted from the electrode using Tris-HCl solution, resulting in the formation of the MIP-modified electrode. For preservation and future testing, it was crucial to store the prepared electrodes in a 2.5 wt.% CaCl_2_ aqueous solution.

### 4.4. Characterizations

The CaAlg, BSA molecularly imprinted CMWCNT/CaAlg (MIP) hydrogel film and NIP film were characterized by a scanning electron micrograph (SEM) and Fourier-transform infrared spectroscopy (FT-IR). The conformation changes of the BSA solution and after elution were characterized by a circular dichromatic spectrum (CD). 

The swelling and mechanical properties of BSA molecularly imprinted CMWCNT/CaAlg hydrogel films were tested. 

The adsorption properties of BSA molecularly imprinted CMWCNT/CaAlg hydrogel films were tested according to the literature [20,22].

The Appendix A contain detailed information on characterizations and measurements. 

### 4.5. Electrochemical Testing

The samples were tested electrochemically using an electrochemical workstation (LANlIKE). The MIP film or NIP film-modified electrode was used as the working electrode, a platinum sheet as the auxiliary electrode, and a bare carbon electrode as the reference electrode. Finally, the bottoms of the three electrodes were immersed in the prepared buffer solution (deionized water: 150 mL, potassium ferricyanide: 0.049 g, potassium ferricyanide: 0.063 g, potassium chloride: 4.473 g), and the tops of the three electrodes were connected to the test apparatus for electrochemical experiments. The differential pulse voltammetry (DPV) experiment was carried out at a scanning rate of 100 mV/s within the potential range of −0.3 V~0.3 V. The pulse amplitude is 50 mV, the pulse width is 50 ms, the pulse period is 0.2 s, and the potential increment is 4 mV.

Solubilization properties of the BSA molecularly imprinted CMWCNT/CaAlg hydrogel film-modified electrochemical sensor were tested, and detailed information is in the Appendix A. 

The linear range and detection limit of the BSA molecularly imprinted sensor were tested according to the literature [32,33]. 

## Figures and Tables

**Figure 1 gels-09-00673-f001:**
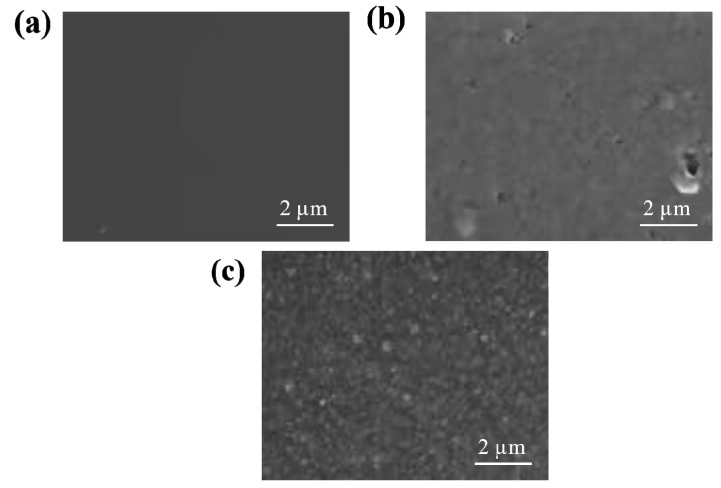
SEM images of the CaAlg hydrogel film (**a**), the BSA-imprinted CMWCNT/CaAlg hydrogel film without eluting (**b**), and the BSA-imprinted CMWCNT/CaAlg hydrogel film after eluting (**c**).

**Figure 2 gels-09-00673-f002:**
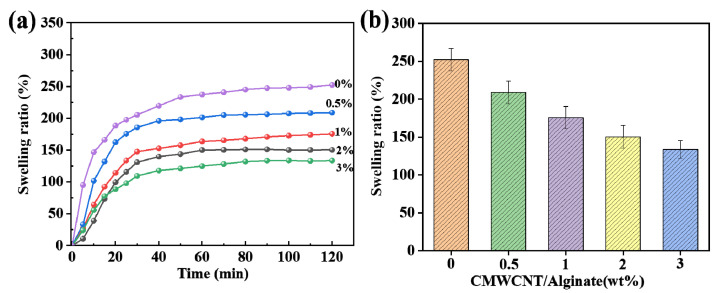
The swelling rate of the CMWCNT/CaAlg hydrogel film in normal saline (0.9 wt.% NaCl) at different times (**a**) and the equilibrium swelling rate of the films with different CMWCNT contents (2 h) (**b**).

**Figure 3 gels-09-00673-f003:**
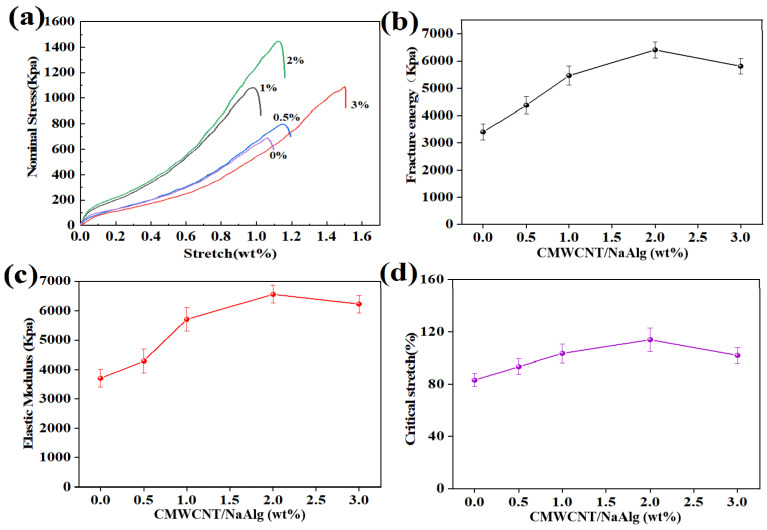
The mechanical properties of BSA molecularly imprinted CMWCNT/CaAlg hydrogel films with different CMWCNT contents. The stress–strain curves of BSA molecularly imprinted CMWCNT/CaAlg hydrogel films with different CMWCNT contents (**a**); the relationship curves between fracture energy (**b**), elastic modulus (**c**), critical stretch (**d**) and CMWCNT content.

**Figure 4 gels-09-00673-f004:**
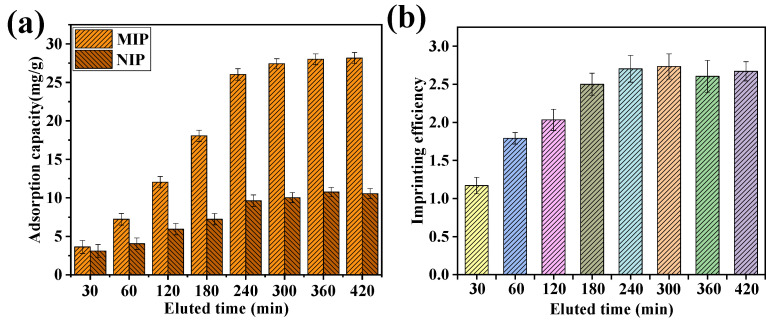
The effect of elution time on the adsorption of MIP and NIP film. The relationship between adsorption capacity and the elution time (**a**), the imprinting efficiency, and the elution time (**b**).

**Figure 5 gels-09-00673-f005:**
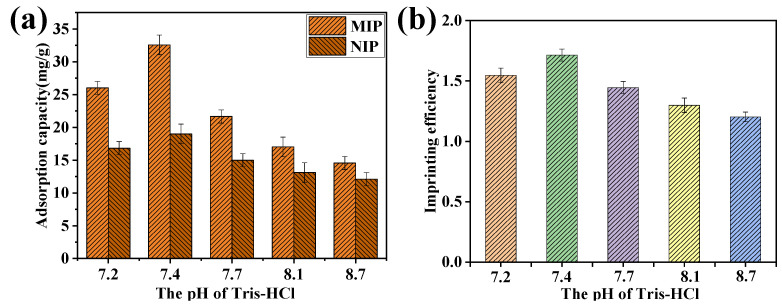
The effect of elution pH on the adsorption of MIP and NIP film. The relationship between adsorption capacity and the elution pH (**a**), the imprinting efficiency, and the elution pH (**b**).

**Figure 6 gels-09-00673-f006:**
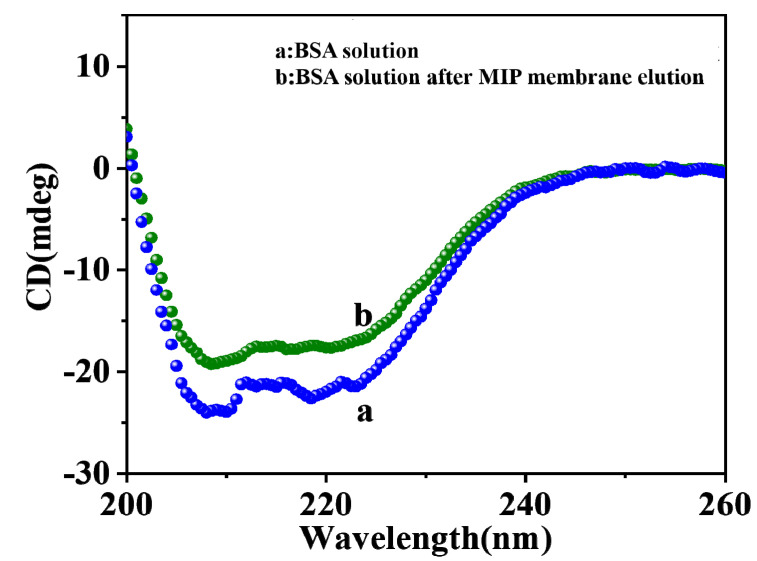
CD spectra of pure BSA (a) and BSA eluted (b) with Tris HCl solution form CMWCNT/CaAlg films.

**Figure 7 gels-09-00673-f007:**
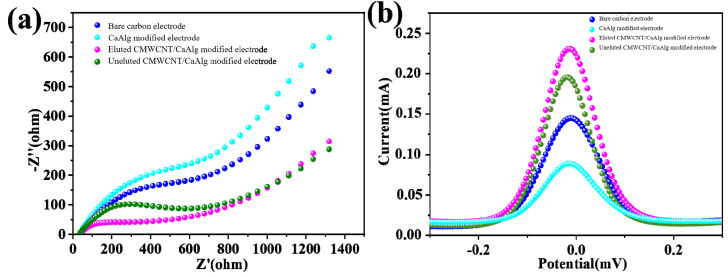
The electrochemical impedance spectroscopy (EIS) (**a**) and differential pulse voltammetry (DPV) (**b**) of the electrochemical sensor modified BSA molecularly imprinted CMWCNT/CaAlg films.

**Figure 8 gels-09-00673-f008:**
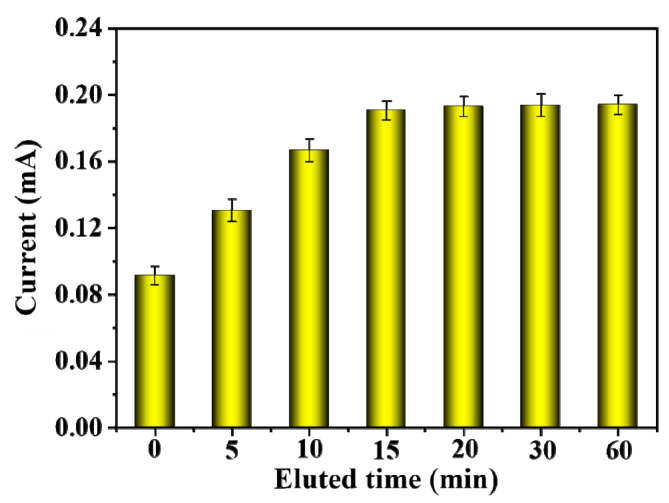
The effect of elution time on the DPV response currents of MIP-modified electrodes.

**Figure 9 gels-09-00673-f009:**
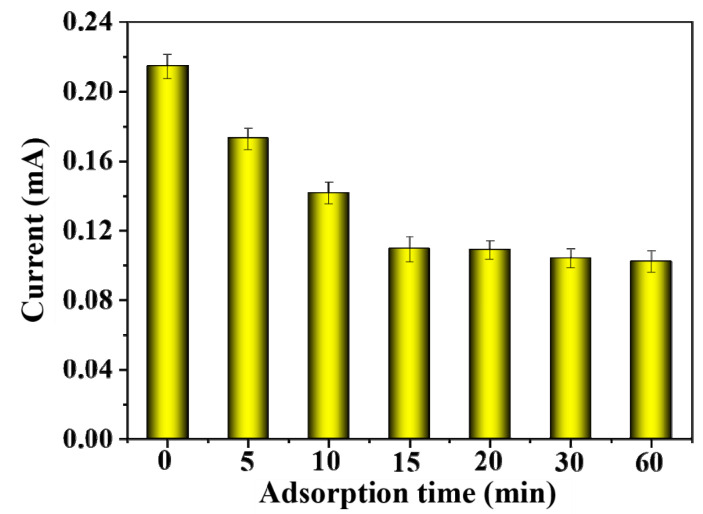
The effect of adsorption time on the DPV response currents of MIP-modified electrodes.

**Figure 10 gels-09-00673-f010:**
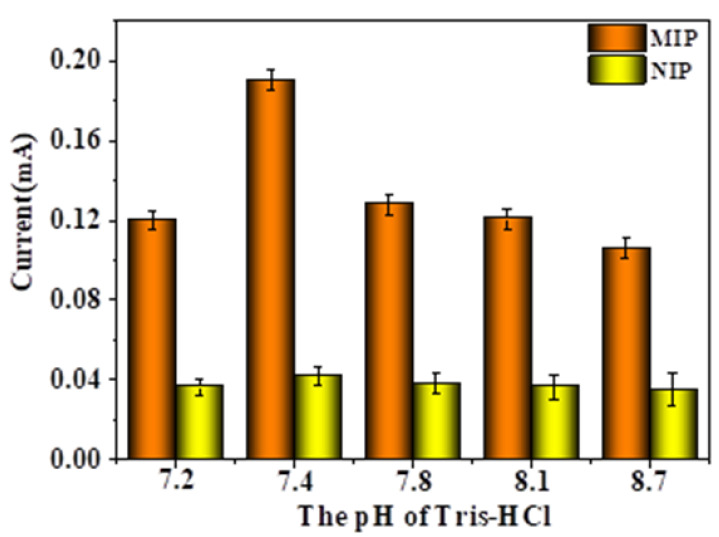
The effect of elution pH on the DPV response currents of MIP- and NIP-modified electrodes.

**Figure 11 gels-09-00673-f011:**
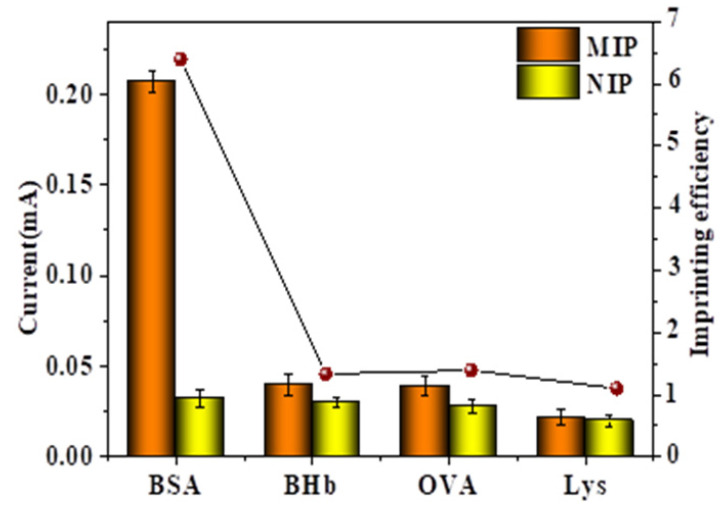
The DPV current response and IF values obtained by MIP- and NIP-modified electrodes adsorbing different proteins.

**Figure 12 gels-09-00673-f012:**
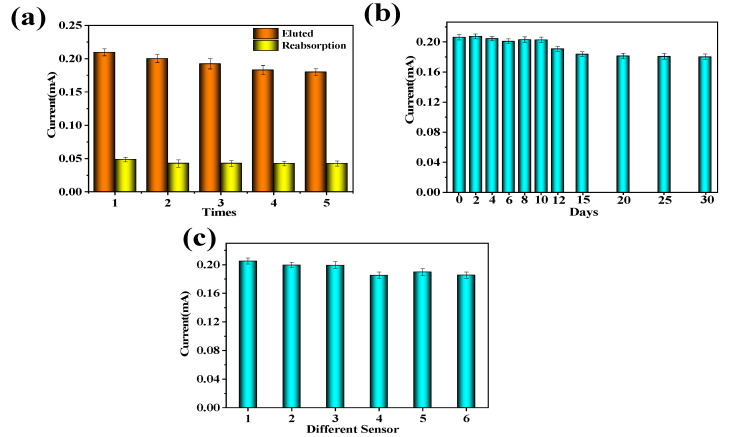
Electrochemical response current of MIP-modified electrode DPV: (**a**) The peak current value of DPV after five cycles of elution and readsorption; (**b**) the peak current value of DPV within 30 days of the same MIP electrochemical sensor; (**c**) DPV peak current values obtained after eluting template molecules with six different MIP electrochemical sensors.

**Figure 13 gels-09-00673-f013:**
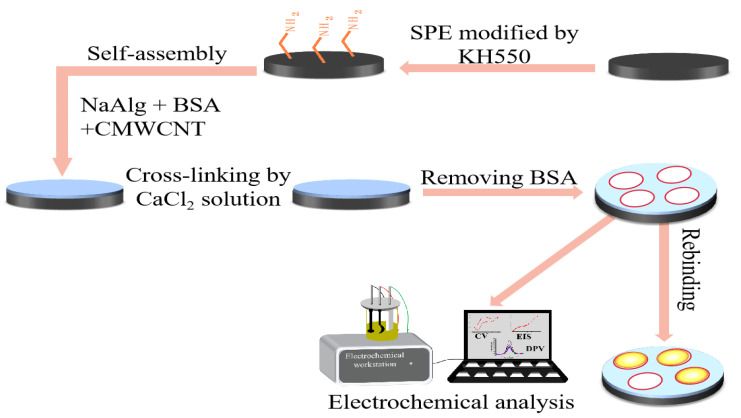
Schematic diagram of the preparation process of a BSA molecularly imprinted CMWCNT/CaAlg hydrogel-modified electrochemical sensor.

**Table 1 gels-09-00673-t001:** Comparison of the performance of the MIP sensors in this work with other materials.

Materials	Linear Range (mg mL^−1^)	Detection Limit (mg mL^−1^)	References
MIP microspheres	1.0 × 10^−5^–5.0 × 10^−3^	1.5 × 10^−6^	[32]
MIP/Cd Te quantum dots	3.3 × 10^−2^–0.66	1.0 × 10^−3^	[33]
Carbon dot	2.0 × 10^−2^–0.1	8.5 × 10^−4^	[34]
CaAlg/CMWCNT hydrogel MIP	1.0 × 10^−6^–1.15 × 10^−3^	5.6 × 10^−6^	This method

## Data Availability

Not applicable.

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
