# Peer review of "Bovine Serum Albumin Molecularly Imprinted Electrochemical Sensors Modified by Carboxylated Multi-Walled Carbon Nanotubes/CaAlg Hydrogels"

_gels, 2023, doi:10.3390/gels9080673_

Round 1
Reviewer 1 Report
In this article, bovine serum albumin (BSA) molecularly imprinted carboxylated multi wall carbon nanotubes (CMWCNT)/ calcium alginate (CaAlg) hydrogel film (MIP) and non-imprinted (NIP) hydrogel electrochemical sensors were studied by SEM and FTIR, as well as characterized the swelling rate, mechanical properties, adsorption properties, circular dichroism, and electrochemical properties. The adsorption capacity of the MIP film for BSA was 27.23 mg/g and the imprinting efficiency reached 2.73 with good selectivity, reusability, stability, and reproducibility. In my opinion, this manuscript, which is well-written and organized, should be accepted for publication in Gels journal after addressing the following:
1) In page 1 line 17, DPV should refer to the differential pulse voltammetry.
2) In page 3, the IR analysis shows no change in the spectra for the CaAlg, NIP or MIP films. I think this study does not provide relevant information thus it might get removed from the paper.
Good enough
Author Response
Comment 1: In page 1 line 17, DPV should refer to the differential pulse voltammetry.
Response: Thank you for your correction. The full name of DPV has been changed to differential pulse voltammetry.
Comment 2: In page 3, the IR analysis shows no change in the spectra for the CaAlg, NIP or MIP films. I think this study does not provide relevant information thus it might get removed from the paper.
Response: Thank you for your advice. The content related to infrared analysis in the article was deleted.
Reviewer 2 Report
This article is comprehensive, logically organized, and contains valuable information on the BSA molecularly imprinted electrochemical sensor modified by carboxylated multi-walled carbon nanotubes/CaAlg hydrogel.
To improve the manuscript, the authors should take the following considerations:
(1) The authors presented the swelling rate of CMWCNT/CaAlg hydrogel film in normal saline (0.9 wt.% NaCl) at different times (a), and the equilibrium swelling rate of the films with different CMWCNT contents (2 h) (b) in Figure 3. The authors should present the cross-linking density of the films with different CMWCNT contents (2 h).
(2) The authors presented the mechanical properties of BSA molecularly imprinted CMWCNT/CaAlg hydrogel film with different CMWCNT contents in Figure 4. The authors should write “CMWCNT/Alginate (wt%)” instead of “Alginate (wt%)” as presented in Figures 4 (b) and (c), for uniformity.
(3) The authors presented the “2.6. Effect of eluent times, adsorption times and eluent pH on DPV response currents of the MIPs modified electrodes” however, the authors should present the mathematical model to verify the experimental results.
The submitted manuscript has significant scientific insights and the conclusions are soundly supported by the experimental data. However, the manuscript requires minor revisions before being accepted in the Special Issue: Alginate-Based Gels: Preparation, Characterization and Application in the well-circulated journal, Gels.
-
Author Response
Comment 1: The authors presented the swelling rate of CMWCNT/CaAlg hydrogel film in normal saline (0.9 wt.% NaCl) at different times (a), and the equilibrium swelling rate of the films with different CMWCNT contents (2 h) (b) in Figure 3. The authors should present the cross-linking density of the films with different CMWCNT contents (2 h).
Response: Thank you for your advice. However, it is not easy to obtain the crosslinking density of hydrogel, and most of the test methods are for chemically crosslinked hydrogels. We have tried to test the cross-linking density of the CMWCNT/CaAlg hydrogel films with different CMWCNT contents according to Flory-Rether equation in the literature [1].
According to the Flory-Rether equation, the calculation formula of the molecular weight () of the net chain between the cross-linking points of the gel is as follows:
(1),
where is the volume fraction of swelling polymer.
= (2)
and is the density of polymer and solvent respectively, Mb and Mathe mass of polymer before and after swelling respectively, and Vsis the Molar volume of solvent. is the interaction parameter. (3), where (4),
where is the slope of the volume fraction temperature curve.
In addition to the determination of parameters in the above formula, the proportion and molecular weight of segment G and segment M in sodium alginate are also required. To fully obtain these parameters, more characterization is required, which will take several months.
It should be pointed out that although is an important parameter of the molecular structure of cross-linked polymers, the cross-linking network of polymers is not ideal, and there may be the role of interpenetration and entanglement of polymer chains to limit the molecular conformation entropy; Alternatively, internal crosslinking occurs on the same polymer chain, generating a closed ring that does not contribute to the elasticity of the interconnection network; In addition, the end of the polymer chain is not fixed at the crosslinking point and does not contribute to elasticity. These factors need to be corrected, but there is currently no more comprehensive theory, so it is only a rough numerical value.
In summary, it seems that it is not worth investing a lot of effort to obtain a less accurate data. In the following research, we will further study the cross-linking density of the CMWCNT/CaAlg hydrogels and analyze them.
[1] Kong H J, Lee K Y, Mooney D J. Nondestructively probing the cross-linking density of polymeric hydrogels. Macromolecules, 2003, 36: 7887-7890
Comment 2: The authors presented the mechanical properties of BSA molecularly imprinted CMWCNT/CaAlg hydrogel film with different CMWCNT contents in Figure 4. The authors should write “CMWCNT/Alginate (wt%)” instead of “Alginate (wt%)” as presented in Figures 4 (b) and (c), for uniformity.
Response: Thanks for your suggestion. “Alginate (wt%)” in Figure 4 has been uniformly changed to “CMWCNT/NaAlg (wt%)”.
Comment 3: The authors presented the “2.6. Effect of eluent times, adsorption times and eluent pH on DPV response currents of the MIPs modified electrodes” however, the authors should present the mathematical model to verify the experimental results.
Response: Thanks for your advice. However, we did not found any mathematical model to verify the experimental results of eluent times, adsorption times or eluent pH on DPV response currents. There were some mathematical models to verify the experimental results of eluent times and adsorption times on the BSA adsorption. The mathematical model of eluent times, adsorption times and eluent pH on the BSA adsorption might verify the experimental results partly because the adsorption capacity of proteins is related to DPV response currents. If we want to delve into the elution and adsorption of CMWCNT/CaAlg membranes at different pH values and conduct mathematical model analysis, it may take several months. Because the time provided by the editorial department is very short, we will fully consider your suggestions in our future research.
In addition, computer simulation methods can also provide a more in-depth and vivid study of protein elution and adsorption processes. We will also conduct molecular dynamics simulations in future research.

Reviewer 3 Report
The topic is fair and the BSA molecularly imprinted electrochemical sensor results show good improvement. However, the manuscript needs to be revised according to the following comments:
(1) A list of all symbols and abbreviations may be useful for readers.
(2) The topic focuses on the electrochemical sensor; therefore the main parameters of this sensor should be described and addressed, e.g. sensitivity, detection limit, and quality factor.
(3) A brief comparison study between the results with previous works is strongly recommended.
(4) Please explain Figure 3(a) in more details.
(5) The references are a bit old for this topic and recent works should be cited.
Author Response
Comment 1: A list of all symbols and abbreviations may be useful for readers.
Response: Thank you for your advice. All symbols and abbreviations have been listed.
Abbreviations
NaAlg sodium alginate
BSA bovine serum albumin
CMWCNT carboxylated multi wall carbon nanotubes
CaAlg calcium alginate
MIP molecularly imprinted polymer
NIP non imprinted polymer
EIS electrochemical impedance spectroscopy
DPV differential pulse voltammetry
CD circular dichroism
Tris three (hydroxymethyl) aminomethane
BHb bovine hemoglobin
OVA ovalbumin
Lys lysozyme
IF imprinting efficiency
ΔI current change value
Comment 2: The topic focuses on the electrochemical sensor; therefore the main parameters of this sensor should be described and addressed, e.g. sensitivity, detection limit, and quality factor.
Response: Thank you for your advice. The main parameters of this sensor was described and addressed, e.g. sensitivity and detection limit. As shown in Fig. S1, the favorable linear relationship between BSA concentration and peak current value in the range of 5.6 × 10-6 mg mL-1 to 1.2×10-3 mg mL-1, with the linear equation: Δ I(μ A) =0.94CBSA+4.21. And the linear correlation coefficient R2 was 0.998. The detection limit (LOD) was 5.6 × 10-6 mg mL-1 (S/N=3).
Fig. S1 Calibration plot between DPV peak current and BSA concentration for MIP electrode.
Comment 3: A brief comparison study between the results with previous works is strongly recommended.
Response: Thank you for your advice. A brief comparison study between the results with previous works was in table 1. It is found that the MIP electrochemical sensor prepared in this work has higher sensitivity and lower detection limits compared with previous reports.
Table 1. Comparison of the performance of the MIPs sensor in this work with other materials.
|
Materials |
Linear range (mg mL-1) |
Detection limit (mg mL-1) |
References |
|
MIP Microspheres |
1.010-5-5.010-3 |
1.5×10-6 |
[32] |
|
MIP/Cd Te quantum dots |
3.310-2-0.66 |
1.0×10-3 |
[33] |
|
Carbon dot |
2.010-2-0.1 |
8.5×10-4 |
[34] |
|
CaAlg/CMWCNT hydrogel MIP |
1.010-6-1.1510-3 |
5.6×10-6 |
This method |
[32] J.H. Yu, F.W. Wan, C.C. Zhang, M. Yan, X.N. Zhang, S.W. Wang. Molecularly imprinted polymeric microspheres for determination of bovine serum albumin based on flow injection chemiluminescence sensor. Biosens. Bioelectron., 26 2010 632-637.
[33] Y. Wang, Q.Z. Hu, T.T. Tian, Y.A. Gao, L. Yu. A nonionic surfactant-decorated liquid crystal sensor for sensitive and selective detection of proteins. Anal. Chim. Acta, 937 2016 119-126.
[34] Y.J. Zhao, Y.J. Chen, M.Y. Fang, Y.B. Tian, G.Y. Bai, K.L. Zhuo. Silanized carbon dot-based thermo-sensitive molecularly imprinted fluorescent sensor for bovine hemoglobin detection. Anal. Bioanal. Chem., 412 2020 5811-5817.
Comment 4: Please explain Figure 3(a) in more details.
Response: Thank you for your advice. Figure 3(a) was explained in more details.
Figure 3(a) shows the stress-strain curves of BSA molecularly imprinted CMWCNT/CaAlg hydrogel film with different CMWCNT contents. The tensile strength of the CMWCNT/CaAlg hydrogel film was higher than the CaAlg hydrogel film. When the CMWCNT content in NaAlg was 2 wt. %, the maximal tensile strength reached with a value of 1440 kPa. Ca2+ cross-linking synchronously with the -COO- of CMWCNT and the -COO- of NaAlg increased the cross-linking density and improved the strength of the CMWCNT/CaAlg hydrogel film. The interactions between CMWCNT and NaAlg were investigated by molecular dynamics (MD) simulation in another paper we published.
Comment 5: The references are a bit old for this topic and recent works should be cited.
Response: Thank you for your advice. Some references were replaced and recent works were cited.
